# Can the Development of Religious and Cultural Tourism Build a Sustainable and Friendly Life and Leisure Environment for the Elderly and Promote Physical and Mental Health?

**DOI:** 10.3390/ijerph182211989

**Published:** 2021-11-15

**Authors:** Hsiao-Hsien Lin, Ko-Hsin Chang, Chih-Hung Tseng, Yueh-Shiu Lee, Chih-Hsiang Hung

**Affiliations:** 1School of Physical Education, Jiaying University, Meizhou 514015, China; chrishome12001@yahoo.com.tw; 2Department of Physical Education, Chinese Culture University, Taipei 111396, Taiwan; zkx2@ulive.pccu.edu.tw; 3Department of Leisure and Recreation Management, Asia University, Taichung 41354, Taiwan; boy217010@hotmail.com; 4General Education Center, National Penghu University of Science and Technology, Kaohsiung 88000, Taiwan; leeys@gms.npu.edu.tw; 5Center for General Education, Taipei Medical University, Taipei 11031, Taiwan

**Keywords:** religious culture, senior citizens, leisure satisfaction, physical and mental health, development balance

## Abstract

From the perspective of satisfaction, physical and mental health, and re-travel, this study explored whether the development of religious and cultural tourism could construct a sustainable and friendly life and leisure environment for the elderly to promote physical and mental health. This research adopted a mixed method, collected 700 questionnaires, used SPSS 22.0 statistical software, and analyzed basic statistics, *t*-test, and PPMCC test. Then, the researchers conducted semi-structured interviews, collected the opinions of six interviewees, and finally analyzed with multiple checks Law discussion. The results found that people of different genders and stakeholders had different opinions about DIY activities on leisure satisfaction, featured itineraries, relaxation areas, signs and instructions, community association and service center services, historical landmarks, and public transportation. They found people’s life satisfaction in physical and mental health was increased, their headaches or pressures on the top of their heads were relieved, backache problems were reduced, and they were no longer anxious and lost tempers. They had a greater willingness to revisit some places and share experiences. This study found significant differences among these topics (*p* < 0.01). Women, residents, and tourists had different opinions. In addition, although the natural environment landscape and feelings have the greatest influence, the better the physical and mental health was improved, the better the willingness to travel. However, the more perfect the local construction and development, the less favorable to attract people to engage in leisure activities or tourism consumption.

## 1. Introduction

In recent decades, there has been an advancement in medical technology, an improvement in social welfare, global economic development, an improvement in living standards, a gradual improvement in sanitary conditions, a reduction in human infection and even the death rate, and an increase in the average life span [1,2]. Although human beings can live longer, the birth rate has been declining year by year. The imbalance between the middle-aged and newborn populations has led to the emergence of an aging social phenomenon in the population structure of various countries [3,4].

Old age refers to people over 65 years old. When the elderly population accounts for 7% of the total population, it is an aging society phenomenon [5]. The phenomenon of an aging society has both positive and negative effects on the country’s overall social and economic development [6]. Due to the increase in age, the knowledge, expertise, and experience accumulated by the elderly, such as the experience, knowledge, technology, personal wealth, and interpersonal interaction needed for life, are extraordinary achievements, and they are one of the most significant populations of all ages in society [7]. However, the body’s self-healing function and immune function decline due to aging, and the body is more susceptible to diseases. It takes a long time to recover and consumes more medical resources [8]. If elderly people are unfortunately infected, treatment will take a lot of time [9]. In addition to the outbreak of COVID-19, the world is at risk of infection, and the virus can be transmitted through multiple routes and the risk of infection is high [10]. Elderly people are more likely to be infected and die than individuals of other age groups [11,12], plus the rate of the second dose of human vaccines in the world is low, and there are even countries where there are still no vaccines [13]. This leads to higher social and economic resource investment for society and the elderly. The scale and quality of the investment will affect the overall social development quality of the country and the physical and mental health of the elderly [9,14,15]. Therefore, it will be an important issue for all countries to assist the elderly in finding ways to improve their quality of life, maintain their physical and mental health, reduce medical resources and expenditures, and provide a stable economy for national development.

Although the development of science and technology and the improvement of the economic level will not change people’s goal of improving the quality of life and soul, the elderly’s participation in leisure and tourism activities can improve physical and mental health [16,17,18,19]. For the elderly, improving the quality of life and the environment and enhancing positive feelings of the soul are two of the key factors to maintaining the satisfaction of the elderly with the quality of life and the environment [20,21,22]. People will lose a lot of role-playing and functions in society due to aging [15]. Although material requirements are reduced, time and quality requirements are higher [23,24]. Especially on the spiritual level, elderly people are one of the more vulnerable age groups. Although they want more mental and spiritual comfort than young people, they do not want to spend too much time [15,20,21,22,23,24]. Therefore, this research focuses on finding ways to improve mental and spiritual quality in a short time and to efficiently improve the physical and mental health of the elderly.

Culture is a state or habit of the soul, and religious culture is one of the ways to comfort the soul [25]. Religious culture originates from people’s anxiety when they are unable to predict, explain, or solve issues scientifically, such as facing wars, natural disasters, etc. There are people, events, and aftermaths that can solve most people’s problems in the current society. As a result, people have sustenance, which has been passed down to form myths or ceremonies and become people’s folk culture, organizations, or life characteristics [26]. Since their evolution, religious and cultural activities have become one of the ways for people to comfort their souls, maintain their beliefs, stabilize personal anxiety, and maintain physical and mental health and quality of life [24,25,26,27]. Because religious culture is supernatural, it cannot explain the attraction of worship and belief forces [25], and religious sacred places, historical sites, or landscapes have long historical and cultural value, which can be used by the public for viewing, investigating, participating, and learning [28]. Therefore, religious and cultural activities can not only provide the elderly with a stable mind and help them maintain physical and mental health [29], but they can also become a resource with characteristics of culture, history, leisure, and tourism [25]. Religious culture is an activity and a characteristic tourism resource. Engaging in religious and cultural activities can help people stabilize their emotions and use resources to plan leisure and tourism activities [25].

Moreover, it can enhance local development and construct a friendly life and leisure environment. Eventually, it can become a plan for the elderly to improve their quality of life and improve their physical and mental health. It is a friendly, safe, and comfortable option.

Fengtian Temple is in Xingang Township, Taiwan. The temple is centered around Mazu. It was started in 1622, rebuilt in 1700, and completed in 1811. It has a history of 399 years. Although it has experienced wars, earthquakes, and other damages in the past, under the follow-up reconstruction and development and the inheritance of existing cultures, it has also become one of Taiwan’s world-renowned Mazu cultural tourism activities to visit [30]. Before the pandemic affected Taiwan, it set a record attracting 9.96 million visitors a year [31,32]. The religious and cultural activities of Fengtian Temple in Xingang have been developed to date to become a well-known local coordinate and tourist destination and one of the main beliefs of residents. It has also become one of Taiwan’s world-famous cultural and tourist attractions [31,32,33,34]. The development of local religious and cultural tourism activities in Fengtian Temple of Xingang has achieved excellent results. 

Although religious beliefs and culture are unique local cultures, they are uniquely attractive [28], which can arouse people’s interest to experience or participate [25]. Older people can relax and stabilize the mind by participating in the process [29] and finally achieve the goal of physical and mental health [16,17,18,19]. In addition, the unique attraction of religious and cultural activities and the effect of calming the soul enable participants to obtain the effects of stabilizing the mind and regulating physical and mental health [28,29]. The resulting leisure, tourism, and economic benefits can provide tourists with viewing, consumption, and experience [35,36] and bring economic benefits to residents, improve the quality of life, and promote physical and mental health [37,38,39,40].

Some scholars believe that although this effect can be compared and verified by the parties involved in the activity process, as well as the expected expectations of the surrounding environment and the actual experience after participation [41], the greater the degree of positive perception, the better the effect and the greater the willingness to consume—the greater [42] and the stronger the willingness to continue participating in the future experience [43]. However, it takes time to verify the effectiveness of decision-making, and we must first obtain the opinions of tourists and residents [25,44], and then verify and compare the two [25,44,45] to obtain more in-depth answers. Furthermore, the current research on the elderly in religious activities or cultural tourism focuses on religious culture [14], economic, social, and environmental benefits [17,26,28,30,36]. The research is based on the influence of people’s experience on the quality of life [21,22,39,40] and physical health [29]. Although there are a few studies on satisfaction and physical and mental health issues [25], they did not analyze the relationship between satisfaction, physical and mental health, and willingness to re-participate at the same time, nor did they discuss the development of religious and cultural activities for the construction of friendship for the elderly, nor how the leisure environment promotes the influence of physical and mental health.

Therefore, the main purpose of this research is to use religious and cultural activities as the theme from the satisfaction of residents and tourists and the cognition of issues such as physical and mental health and willingness to travel again to explore whether the development of religious and cultural activities can build a sustainable and friendly leisure environment for the elderly and promote physical and mental health. The results of this research are expected to be used by the government to plan better physical and mental health care measures for the elderly after the pandemic is over and alternative cultural resource development and utilization decisions. Moreover, the study could provide suggestions for older people to plan leisure and tourism activities in the future. They can simultaneously obtain the benefits of improving the quality of life, calming the soul, and improving physical and mental health.

## 2. Literature Discussion

### 2.1. Cultural Activities Are Important for New Students Who Are Accustomed to the Leisure Environment and Improve Their Physical Health

With the outbreak of COVID-19, people’s lives and activities are limited, and as people grow older, their body’s self-healing function and immune function gradually decline [8]. This causes widespread psychological distress, mental illness, physical pain, and other symptoms [46]. In particular, elderly people have weak self-prevention and immune functions and are vulnerable to diseases [9]. However, some scholars say that after the elderly engage in leisure or tourism activities, not only can they improve their personal health [25], they can also fill up their empty hearts and relieve stress by participating in cultural activities [24,25,26,27] and finally achieve a good body and mind and health status [24,25,26,27,28,29]. However, studies have confirmed that in a pandemic environment, the psychological stress of the elderly is more than 1.5 times that of other age groups [47], and the risk of infection is higher [9]. Therefore, if this problem is not resolved, it will impact the current national economy, industry, and social welfare development plans.

Religious belief is one of the cultures of mankind, which can stabilize people’s hearts and become an invisible force for social stability, constructing a comfortable and stable living space for people [24,25,26,27]. In addition to the development of religious culture, the characteristics of local characters, languages, food, drama, music, dance, crafts, architecture, and festivals are derived [47]. Therefore, religious culture can make all people who feel anxious due to the pandemic stabilize their emotions and achieve comfort in their hearts. It can also allow tourists to experience the characteristics of different religions and cultures. At the same time, it can better enable residents to obtain a stable income, improve economic difficulties, and improve their life quality [25,48].

Furthermore, although the physical and mental health of the elderly is easily affected by diseases, if the elderly can be provided a safe and comfortable environment, they can engage in leisure and tourism activities [20,21,22] so that they can calm their minds and relax their minds and bodies [15,20,21,22,23,24]. This will help maintain physical and mental health, reduce the risk of infection and life pressure, and reduce physical and mental harm [16,17,18,19]. Therefore, the research believes that when the current society faces pandemics and economic shrinkage, religious culture is the core, and religious culture or tourism resources are used to construct a friendly leisure environment. Providing leisure or tourism activities for the elderly will help stabilize the anxiety of the elderly, maintain their physical and mental health, and improve their current difficulties.

### 2.2. Revisit Intention

Willingness is an individual’s prediction of whether to perform future behaviors [49]. It is also a willingness to anticipate or plan future behavior, which can determine the possibility of the implementation of the plan [50]. The willingness to travel refers to the frequency and type of potential purchase behavior of individuals participating in tourism activities and the possibility of tourists going to a certain tourist destination [51,52].

The main determinant of the willingness to travel is the tourist’s subjective image of the destination, which can also be regarded as a consumption behavior [53]. Due to the decline in health and mobility of the elderly, there are many considerations for leisure or tourism [20,21,22]. Therefore, the elderly usually set the important attributes of various tourism destinations according to their preferences and then select several personal preference attributes as the evaluation criteria for choosing a destination for leisure or tourism [54]. Of course, it will also affect the willingness to travel to the destination again because of the image or the feeling of the actual experience effect [55]. Therefore, the willingness to travel can be regarded as the expression of loyalty and can also be regarded as the local community’s development decision-making and living environment planning effect to construct a safe and friendly leisure space that allows people to be willing to travel or experience again and be trusted by the public [41,42]. Additionally, when people’s positive feelings about the environmental experience are higher, the awareness of going to travel will be stronger. This will help people make decisions such as revisiting or re-participating in the event or place, simply sharing experience, and being willing to take the initiative to recruit other people to consume. Additionally, different background cognitions will produce differences [52,53,54,55].

From topics such as going to travel, simply sharing experience, or being willing to take the initiative to recruit other people to consume together, the theory of tourism willingness can be used to analyze the degree of identity of tourists to the effectiveness of local tourism development. This research believes that it can also be used to understand whether religious and cultural activities can be the main axis of development and whether a friendly and safe leisure environment can be constructed, recognized by the elderly, and improve their physical and mental health.

### 2.3. Leisure Satisfaction and Revisit Intention

Leisure satisfaction is the positive perception people feel after being aware of their leisure experiences and the environment when they engage in leisure activities [56,57]. To put it simply, people evaluate the degree of satisfaction of individual needs obtained during the leisure experience process, which is leisure satisfaction.

Leisure satisfaction is the use of personal activity experience to compare previous experience, personal expectations, or sense of expectation [58]. When the actual leisure environment or activity content meets personal expectations, a satisfying attitude emerges in the heart [57,58]. The higher the satisfaction level, the higher the positive energy cognition obtained [59]. The more satisfying personal psychological needs can be obtained, the more satisfied with the current life and feelings and the willingness to revisit [60,61]. Some scholars believe that although the influencing factors of leisure satisfaction can be divided into internal and external interference [62], it can be analyzed from six levels: physiology, psychology, education, aesthetics, social interaction, and relaxation [63]. It can also be discussed by society, psychology, aesthetics, education, relaxation, etc. [64]. Among them, the content of leisure activities, enhancing self-confidence and physical fitness, restoring physical and mental health and physical strength, increasing knowledge and social interaction, self-exploration, and relaxation, experience, and feelings toward the environment can be obtained in more in-depth discussions. Additionally, research has pointed out that different background cognitions will produce differences [65,66].

Therefore, if the leisure satisfaction theory is adopted, it can be classified into physical, psychological, educational, aesthetic, social, and relaxing levels. It can analyze the topics of leisure activities, improving self-confidence and physical fitness, restoring physical and mental health and physical strength, increasing knowledge and social interaction, self-exploration, relaxation, environmental experience, and feelings, etc. It can also be discussed from the perspective of different ages. If the community takes religious culture as the core of its development, the constructed environment can make the elderly feel a safe and friendly leisure environment and generate the idea of re-travel.

### 2.4. Physical and Mental Health and Revisit Intention

Personal physical and mental health refers to physical, psychological, and social well-being [58]. It is an analytical method of self-perception evaluation [59], which can present the actual situation through scientific test evidence such as self-assessment [60]. The higher the health risk, the greater the influence on individual behavioral decisions [10,28].

According to related literature, investigating individuals’ physical and mental health based on personal feelings can show the impact of the current environment on people [10]. Physical and mental health can be divided into three levels: psychology, spirit, and attitude [28,61,62]. It is confirmed by feelings such as anxiety, ability, enthusiasm, headache, abdominal pain, insomnia, stomachache, abnormal diet, and the idea of seeking death [63,64]. The current state of physical and mental health will affect the individual’s willingness to act and make judgments, and there will be differences in cognition from different backgrounds [6,28].

Therefore, if the physical and mental health self-checking table is used to evaluate anxiety, ability, enthusiasm, headache, abdominal pain, insomnia, stomach pain, abnormal diet, and death-seeking ideas from three levels: psychology, spirit, and attitude, it can be understood that if the community takes religious culture as the core of development, the environmental atmosphere constructed can improve the physical and mental health of the elderly and generate the idea of re-travel.

## 3. Methods

### 3.1. Research Process and Framework

According to the literature, culture is a habit [25], religious culture has the effect of calming the soul [24,25,26,27], and leisure activities and tourism behaviors both have the effect of improving physical and mental health [28,29]. If religious culture is the core to construct a leisure environment and promote the development of local tourism, both residents and tourists should be able to obtain benefits [37,38,39,40]. Therefore, the framework of this research is based on the research scope of Fengtian Temple in Xingang, Taiwan. It adopts a mixed research method to collect the views of residents and tourists from the aspects of satisfaction, physical and mental health, and re-travel. It investigates whether to develop religious and cultural activities. It can build a sustainable and friendly leisure environment for the elderly and promote physical and mental health. The research framework is shown in Figure 1.

Although the current religious activities, cultural tourism, and leisure environment experience have diverse research directions for the elderly [14,17,21,22,25,26,28,29,30,39,40], few have targeted the elderly and analyzed the relationship between leisure satisfaction, physical and mental health, and willingness to re-participate in the religious–cultural environment at the same time. Therefore, mixed research methods were used, quantitative research methods were first used to supplement the research breadth [67,68], then qualitative research was used to increase the depth of research [69], and multivariate verification methods were used to discuss [70,71,72], which could make up for research development or defects of insufficient theory [73]. Therefore, the research first referred to the compiled questionnaire tools in the literature on leisure satisfaction [25,44,45,57,58,59,60,61,62,63,64,65,66], physical and mental health, [25,44,45,58,59,60,61,62], and willingness to participate again [25,44,45,50,51,52,53,54,55]. First, three experts checked the content validity. After the draft was revised, 100 questionnaires were collected in June 2020, analyzed using SPSS 26.0 statistical software, and confirmed with the formal questionnaire tools.

During the formal questionnaire survey, due to the limitation of the research topic, the interviewees had to first indicate that they were residents or tourists with tourism experience before the survey, and they had a certain degree of understanding of the local development situation before they were the sampling objects. Furthermore, most of the interviewees were older people. Some needed long-term assisted reading or oral surveys due to literacy, eyesight, and the operation of the online questionnaire platform. In addition to the limitation of factors such as the pandemic period, it was precisely the questionnaire collection time from July 2020 to January 2021, which lasted for seven months. A total of 800 questionnaires were issued, 700 copies were effectively recovered, and the recovery rate was 87.5%. The questionnaire data were analyzed with a basic statistics *t*-test and PPMCC test. Then, a semi-open interview method was used to collect the opinions of officials, experts, and residents based on the results of the questionnaire analysis. Then, all the data were assembled in a rigorous, sequential, and logical manner and then valuable information was summarized through summarizing, organizing, and collating methods [70]. Finally, triangulation conducted the discussion in a multi-data and multi-perspective manner [71,72].

### 3.2. Hypothesis

Based on the literature description and the framework diagram mentioned above, this study put forward five hypotheses and arguments, which would be explained as follows.

**Hypothesis** **1.**
*Assume that people’s perception of leisure satisfaction would be consistent.*


Decision-making and development are expected to improve the current situation of local development, construct a friendly leisure environment, and provide a safe and convenient tourism space, which ultimately needs to be experienced and verified by people [25,44,45]. Although it is expected that there would be a gap between the results of decision-making and actual experience, the use of religious culture to conduct rural construction should improve local predicaments, provide people with a stable and comfortable living environment, and promote health [37,38,39,40]. Therefore, the researchers hypothesized that people’s perception of leisure satisfaction would be consistent.

**Hypothesis** **2.**
*Assume that people’s perceptions of physical and mental health would be consistent.*


Decision-making and planning can improve local economic development difficulties, increase employment opportunities, and plan a good leisure environment simultaneously [25,44]. So, people can achieve the goals of obtaining a safe and relaxed living environment, relieving stress, improving quality of life, and maintaining physical and mental health [25,29]. Therefore, the researchers hypothesized that people’s perceptions of physical and mental health would be consistent.

**Hypothesis** **3.**
*Assume that people’s perception of re-travel willingness would be consistent.*


Although there might be a gap between the expected decision-making and the experience, religious culture is a unique local life custom. It has the intangible benefit of calming the soul [28,29]. Therefore, when people face unexpected crises, the living environment constructed with religious and cultural resources as the core would become the main reason people live and travel for consumption [24,25,26,27,28]. Therefore, the researchers hypothesized that people’s perceptions of re-travel intentions would be consistent.

**Hypothesis** **4.**
*Assume that leisure satisfaction and perception of re-travel willingness would have a positive and significant effect.*


Leisure satisfaction is an evaluation formed by comparing personal expectations or a sense of anticipation after using personal experience of the surrounding environment, equipment, and facilities [57,58]. The higher the satisfaction level, the greater the positive energy cognition obtained [59], and the stronger the intention to increase the willingness to revisit [60,61]. Therefore, the researchers assumed that the public would have a positive and significant impact on leisure satisfaction and re-travel willingness.

**Hypothesis** **5.**
*Assume that physical and mental health and the perception of re-travel willingness would have a positive and significant impact.*


Physical and mental health means that when people face the surrounding environment or things, the individual’s physical, psychological, and social levels are sensitive [58], which affects physical and mental health [10]. The higher the health risk, the greater the influence of individual behavioral decisions [10,28]. Therefore, the researchers assumed that people would positively and significantly impact their physical and mental health and perception of re-travel.

### 3.3. Research Tool Design and Analysis

The research adopted different research methods. First, we adopted quantitative research to investigate. The structure and content of the questionnaire were compiled regarding related literature such as leisure satisfaction [57,58,59,60,61,62,63,64,65,66], physical and mental health [58,59,60,61,62], and re-travel willingness [50,51,52,53,54,55]. The questionnaire was divided into two parts. The first was the sample background, which included physical distinction (residents, tourists), gender (male, female), etc. The variable parts included leisure satisfaction, physical and mental health, and re-travel willingness. The questionnaire used a Likert 5-point scale; 1 point meant very dissatisfied, and 5 points meant very satisfied. After editing, three experts reviewed the questionnaire. Then, the researchers invited six interviewees, who were experts with expertise in tourism development decision-making research, residents, tourism operators, and tourists, to put forward their opinions based on the results of subsequent data analysis. The relevant background was the interview topics, as shown in Table 1.

After determining the issues, SPSS 22.0 statistical software and statistical verification measurement were used. The result of the Kaiser–Meyer–Olkin test (KMO) was >0.06 and the *p* value in the Bartlett test was less than 0.01 (*p* < 0.01); this indicated that the scale was suitable for continuous factor analysis [74]. The a was greater than 0.60, which meant that the issue with good reliability [75] and could be analyzed continuously.

Referring to the literature [57,58,59,60,61,62,63,64,65,66], the leisure satisfaction questionnaire was edited and designed with a total of 30 questions. The analysis showed that KMO was 0.980, while Bartlett’s approximate χ^2^ value was 14,409.01, df was 435, and the significance was *p* < 0.001, which was suitable for factor analysis. The explanatory variance of the scale was 14.588%, 11.765%, 9.195%, 8.643%, 7.104%, 5.291%, and 3.37%, and the total explained variance was 59.957%. After factor analysis, good and reliable topics were retained. These topics were named as natural environment and landscape (4 questions), history, culture and humanities (4 questions), special products (3 questions), service facilities (4 questions), space capacity (4 questions), popularity (4 questions), tourism information and safety (3 questions), and price (4 questions), among which the α coefficient was 0.969–0.970. The total table α coefficient was 0.971. Based on the above analysis results, this questionnaire had good reliability.

Referring to the literature [58,59,60,61,62], the physical and mental health questionnaire was edited and designed with nine questions. The analysis result showed that KMO was 0.723, while Bartlett’s approximate χ^2^ value was 1416.34, df was 3, and the significance was *p* < 0.001, which was suitable for factor analysis. The explanatory variance of the scale was 75.883%, and the total explanatory variance was 75.883%. After factor analysis, good and reliable topics were retained. The topics were named as mental feelings (3 questions), mental state (3 questions), and life attitude and health (3 questions). The α coefficients were 0.970–0.974, and the total scale α coefficient was 0.976. Based on the above analysis results, this questionnaire had good reliability.

Referring to the literature [50,51,52,53,54,55], the questionnaire on re-tourism intention was edited and designed with a total of 3 questions. The analysis result showed that KMO was 0.891, Bartlett’s approximate χ^2^ value was 10,021.87, df was 36, and the significance was *p* < 0.001, which was suitable for factor analysis. The explainable variation of the scale was 42.767%, 37.906%, and 9.549%, and the total explained variation was 89.921%. After factor analysis, good and reliable topics were retained. They were named as willing to continue living or traveling due to activities, sharing living or traveling experience, willingness to revisit the local area, etc., in which the α coefficient was 0.819–0.900. The total table α coefficient was 0.900. Based on the above analysis results, this questionnaire had good reliability.

### 3.4. Scope and Objects

Mazu is the main belief in Fengtian Temple, located in Xingang Township, Chiayi County, Taiwan. Although the area has experienced wars, earthquakes, and other persecutions, the local government has combined religious and cultural resources to improve community development, followed by investment in the reconstruction and promotion of unique festivals and folklore activities. It has a history of 399 years. Fengtian Temple has become one of the main gathering places and beliefs of residents in Xingang Township. It has also become a well-known coordinate and tourist destination and a famous cultural and tourist attraction in Taiwan [30,31,32,33,34]. The local area expects to use religious and cultural resources to develop a youth paradise and build a friendly elderly community. It has a total of 8650 older people and attracts a record of 9.96 million people every year [31,32,40,76], which shows the important status of Fengtian Temple in Xingang Township. This can also help us to understand the effectiveness of the local use of religious and cultural resources to develop a friendly leisure life and tourism environment for the elderly.

Therefore, this study took Fengtian Temple as the case, Xingang Township as the scope of the investigation, residents and senior citizens of tourists as sample collection objects, and used theories of leisure satisfaction, physical and mental health, and re-travel willingness as topics. Exploring whether religious and cultural activities could build a sustainable and friendly leisure environment for the elderly and the topic of promoting physical and mental health was representative.

### 3.5. Methods and Limitations

This research used mixed research methods. However, due to the threat of the COVID-19 pandemic and restrictions on funding, manpower, and material resources, when conducting field surveys during the survey period, the convenience sampling method was used for sampling simultaneously. The online questionnaire platform was then supplemented, and the snowball sampling method was used to expand the collection of questionnaires with local people as the main survey subjects. In addition, in the design of interview topics, experts in decision-making analysis, tourism, leisure/and other fields, local senior citizens, and businesspeople were invited to use video systems and telephones to obtain the interviewee’s consent to interview willingness. The survey was conducted in a semi-structured interview mode, and opinions were put forward on the analysis and results of the questionnaire. In the end, all the data were collected and discussed using multiple verification methods.

However, as explained above, due to the risk of infection and research methods and restrictions during the investigation period, the research might produce differences in the number of research samples or results. The defects would become research recommendations. The researchers would look forward to follow-up investigators to investigate and improve the deficiencies.

### 3.6. Ethical Considerations

This study set Fengtian Temple as the case, Xingang Township as the scope of the investigation, and senior citizens of residents and tourists as the sample collection objects. This study used theories of leisure satisfaction, physical and mental health, and re-travel willingness as topics. The researchers collected data with mixed research methods and used multivariate verification and analysis methods to discuss. Due to the research topics and sampling restrictions, although the interviewees were older and the collection process might be restricted by factors such as physiology or technical software operation capabilities, they all had a good understanding of the research topics and current conditions. Moreover, before the interview and after the survey, the study repeatedly confirmed the respondents’ willingness to accept and provide data. Therefore, all interviewees agreed with and understood the main idea of this research, recorded and collected data anonymously and knowingly, and agreed to cooperate in the provision of relevant data. Therefore, the research design and data collection process conformed to ethical standards, and there was no need to apply for ethical certification [77,78].

## 4. Analysis and Discussion

The research was explored with 700 valid questionnaires. The analysis showed that, in the sample background, there were 288 residents (41.1%), 412 tourists (58.9%), 412 men (58.9%), and 288 women (41.1%). The basic statistical verification, *t*-test, and Pearson poor performance correlation analysis methods were used to calculate the sample background and leisure satisfaction, physical and mental health, and re-travel willingness.

### 4.1. Leisure Satisfaction Analysis

The *t*-test method, which analyzed the differences in cognition of leisure satisfaction of people with different rights and genders, was used to understand people’s feelings about leisure activities and the living environment space that constructs a religious, cultural atmosphere. The analysis is shown in Table 2. The analysis showed that different genders participated in DIY experience activities (3.49:3.59), special itineraries and activities (3.53:3.36), spacious activities and leisure spaces (3.80:3.49), detailed signs and explanations (3.67:3.51), community associations, and service center consulting services (3.64:3.44). Different rights holders had significant issues on long historical monuments (3.67:3.58), the convenience of public transportation (3.62:3.72), detailed signs and explanations (3.64:3.58), etc. There was a significant difference in these issues (*p* < 0.01). It could be seen that there were different opinions on the development effectiveness of issues such as different genders were concerned with issues such as DIY experience activities, special itineraries and activities, spacious activities and leisure spaces, detailed signs and explanations, community associations, and service center consultation services, different rights holders’ opinions on historical sites, public transportation, and signs. In addition, women paid more attention to DIY experience activities than men, and tourists paid more attention to the convenience of public transportation than residents.

This study showed that the religious culture was a local unique life custom and belief. The pandemic not only brought the elderly at risk of infection but also restricted their range of activities and freedom of movement. However, it would derive characteristics from other areas in folk activities, art, architecture, leisure, and food under long-term development. The characteristics would also become local unique and attractive leisure and tourism activities. Furthermore, Taiwan has had a wealth of experience promoting rural tourism development for a long time, and the plan has been well planned. If the rural areas have been developed with local religious culture as the core, the construction of leisure activities and tourism environment will attract people to live or travel. It is believed that encouraging the elderly to participate in cultural tourism activities after the epidemic is over in the future can be one of the best suggestions for improving the physical and mental health of the elderly. However, rural residents have been enthusiastic and hospitable, and most people have been willing to help tourists solve problems; rural areas are mostly remote. Compared with the quality of public transportation and road planning in urban areas, rural areas have been less convenient for tourists to travel to.

Furthermore, the aging of the local population structure has been unfavorable to promote high-labor leisure or tourism activities, and rural areas are mostly agricultural land and villagers’ settlements. Usable space is limited, and women usually are more interested in hand-made or hands-on activities. Therefore, when people experience the leisure life and tourism environment constructed with religious culture as the core of development, people of different genders and rights holders would have different views on the current development of DIY activities, special itineraries, leisure spaces, signs and explanations, services of community associations and service centers, historical sites, and public transportation. Additionally, DIY activities are favored by women. When traveling in rural areas, tourists pay more attention to the convenience of public transportation.

### 4.2. Cognitive Analysis of Physical and Mental Health

The *t*-test method was used to analyze the differences in the self-cognition assessment of the physical and mental health of people with different rights and genders to understand people’s feelings in the leisure and life environment constructed by the religious–cultural atmosphere. The method is shown in Table 3. Analysis showed that different genders had increased life satisfaction (2.45:2.52), relieved headaches or overhead pressure (2.44:2.64), reduced backache problems (2.28:2.44), and reduced anxiety and loss of temper (2.38:2.54). There were significant differences (*p* < 0.01) between people with different rights in the relief of headaches or overhead pressure (2.55:2.52). It could be seen that different genders had different views on increasing life satisfaction, reducing headaches or overhead pressure, reducing backache problems, no longer being anxious, losing their temper, and people with different rights and interests on issues such as reducing headaches or overhead pressure. Among them, women felt more satisfied with life, had fewer headaches or overhead pressure problems, had fewer backache problems, and no longer felt anxious and lost their temper. Residents felt more strongly for relief of headaches or overhead pressure.

This study showed that culture was a living habit, and religious culture was a kind of belief and folklore. Due to long-term development, it had derived unique language, humanities, art, and architectural characteristics. It had the characteristics of stabilizing people’s hearts, calming emotions, and being rich in leisure activities and tourist attractions, enriching knowledge, technology, and experience. The research team believed that if the elderly could be encouraged to participate in cultural tourism activities, this could accelerate the improvement of the current physical and mental health of the elderly. Moreover, early rural life was difficult, and people’s life was not easy. Men must complete more than one year of military service, experience physical and mental health, and fulfill statutory responsibilities and obligations. Therefore, when the people faced the current social work and life pressure, men resisted the pressure more than women, and it was easier for residents to adapt to the rural living environment.

### 4.3. Analysis of Re-Travel Willingness

The *t*-test method was used to analyze the cognitive differences between gender and people with different rights and their willingness to travel to understand people’s feelings in the leisure and living environment constructed with a religious, cultural atmosphere. The analysis is shown in Table 4. The analysis showed that different genders were willing to revisit the place (2.84:2.87), and people with different rights were willing to share life or travel experience (2.59:2.52) and other issues (*p* < 0.01). Among them, women were willing to visit the place again, and residents were more willing to share life experiences than tourists. It could be seen that different genders were willing to revisit the place, and people with different rights had different views on issues such as willingness to share residential life or travel experience. Among them, women were willing to revisit the place, and residents were more willing to share life experiences than tourists.

Residents and tourists both have ingested or used local resources to meet their personal life and leisure travel purposes. Compared with cities, the countryside has a beautiful natural environment, simple folk customs, a comfortable life, and perfect public security, making it more suitable for living or engaging in leisure and tourism activities. However, transportation planning in rural areas is not convenient. This was the common feeling and expectation of residents and tourists. It is also a topic that needs to be paid attention to in the development of local cultural tourism after the end of the pandemic.

In addition, women are less resistant to stress. If religious culture is used as an example, the development features will be able to relax the body and mind, relieve stress, and stabilize the soul. As a result, women are looking forward to returning to the local area again to relax. Residents are willing to share rural life experiences and provide tourists with references.

### 4.4. Correlation Analysis of Leisure Satisfaction, Physical and Mental Health, and Willingness to Travel Again

Pearson’s poor performance correlation analysis was used to analyze the related effects of leisure satisfaction, physical and mental health, and re-travel willingness. The analysis is shown in Table 5. The analysis showed that leisure satisfaction, physical and mental health perception, and re-travel willingness had significant impacts (*p* < 0.01), negatively correlated with satisfaction. In contrast, physical and mental health were positively correlated. The result was different from research Hypothesis 4, but it met research Hypothesis 5. Among them, the natural environment landscape (−0.200) of the leisure satisfaction dimension had the greatest impact, and the service facility (−0.081) had the least impact; the psychological perception of physical and mental health (0.721) had the greatest impact, and life attitude and health (0.570) had the least impact.

There was a correlation between leisure satisfaction, physical and mental health, and re-visit intention. In addition, the natural environment landscape and mental feeling had the highest influence, and the service facilities, life attitude, and health interference were the least obvious. However, the better the physical and mental health, the higher the willingness to travel again. However, the better the satisfaction, the lower the willingness to swim again.

This study suggested that although tourists expect leisure activities and tourism environments to be equipped with modern facilities and convenient and fast services, leisure activities and tourism environments that are usually too technologically advanced often change the local area’s original appearance and characteristics and lose its local color. However, because the religious culture had a stable mood, engaging in leisure and tourism activities could help relieve stress, improve quality of life, and improve physical and mental health. Therefore, in a leisure and tourism environment where religious culture was the core of development, the effectiveness of developing the local natural environment landscape and personal feelings were the keys to re-tourism. Although the more the physical and mental health was improved, the better the willingness to travel, and the better the effectiveness of the development and planning of local leisure activities, it was not conducive to attracting people to engage in leisure activities or tourism consumption.

## 5. Conclusions

This study found that when older people of different genders and rights holders experienced the development of religious culture and the construction of leisure activities and tourism environments, personal life satisfaction, headache or head pressure, backache, anxiety, irritability, and other physical and mental health problems had obvious effects. Among local DIY activities, special itineraries, leisure spaces, signage and commentary, community associations and service center services, historical sites, public transportation, and other focused developments, women valued DIY activities and looked forward to improving life satisfaction and alleviating headaches, head pressure, and backache. Residents were willing to share life experiences, and tourists hoped to improve the convenience of public transportation. Furthermore, although the natural environment landscape and feelings had the highest influence and the better the physical and mental health was improved, the better the willingness to travel, and the better the local construction and development, it was not conducive to attracting people to engage in leisure activities or tourism consumption.

Suggestions will be provided for government agencies, villages, and follow-up studies.

### 5.1. For Government Agencies

Make good use of the existing humanistic, artistic, food, and architectural features of religious culture. Use the wisdom of senior citizens, cultivate local guides, and commentaries to develop different types of experience activities, attract senior tourists of different genders to participate, and enhance leisure and tourism characteristics.

### 5.2. For the Village

Use the unique folk customs of rural areas and people’s enthusiastic attitude to encourage residents to participate in village development decisions and establish religious–cultural leisure activities and tourism environment characteristics to attract more older people to live there and visit, promote people’s communication and interaction, increase the population, and build and construct warm, friendly, and safe leisure activities and tourism environments for senior citizens to achieve the goal of a paradise for senior citizens.

### 5.3. The Follow-Up Research and Development

The research topic was Fengtian Temple in Xingang Township, Taiwan, and the elderly were the subjects. Due to research topics, manpower, material resources, funds, and the investigation environment, the sample collection method, quantity, and survey objects were limited. Therefore, follow-up researchers can research different countries, religions, cultures, and ages and discuss environmental experience value, happiness, etc. This can improve the current research in high-level fields.

## Figures and Tables

**Figure 1 ijerph-18-11989-f001:**
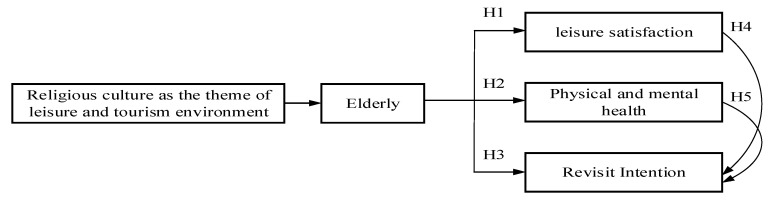
Study framework.

**Table 1 ijerph-18-11989-t001:** Respondents’ background information and an overview of the interview outline.

Identity	Gender	Residence Time/Years of Work Experience	Identity	Gender	Residence Time/Years of Work Experience
Resident	Male	32	Professor	Male	10
Tourist	Female	38	Professor	Female	20
Professor	Male	18	Entrepreneur	Female	32
Construct	Issues				
Impact of tourism development	1. Please give your opinion on the effectiveness of local leisure life and tourism development and briefly explain the reasons. 2. Please provide your opinion and briefly explain the reasons based on your personal physical and mental health perception after engaging in leisure life and tourism activities in the local area. 3. Please provide your opinion and briefly explain the reasons based on your personal willingness to travel after engaging in leisure life and tourism activities in the local area.				

**Table 2 ijerph-18-11989-t002:** Analysis of cognitive differences in leisure satisfaction with different background variables.

Secondary Facets	Issue	Male		Female		*p*	Resident		Tourist		*p*
		M		SD			M		SD		
Natural environment and landscape	Beautiful scenery	3.70	1.202	3.49	1.123	0.470	3.60	1.191	3.63	1.164	0.435
	Special weather patterns	3.55	1.131	3.65	1.123	0.874	3.62	1.189	3.57	1.084	0.151
	Diversified animal and plant resources	3.67	1.171	3.51	1.207	0.339	3.47	1.243	3.70	1.139	0.048
	Comfortable environment	3.55	1.167	3.70	1.135	0.215	3.50	1.189	3.69	1.126	0.234
History, culture, and humanities	Historical site	3.66	1.162	3.56	1.214	0.540	3.67	1.098	3.58	1.241	0.009 *
	Culture museum	3.67	1.065	3.55	1.131	0.116	3.67	1.055	3.58	1.119	0.094
	Rural landscape	3.67	1.145	3.61	1.294	0.015	3.66	1.145	3.64	1.251	0.062
	Special dishes	3.67	1.162	3.44	1.199	0.368	3.62	1.186	3.54	1.180	0.602
Featured products	Featured products	3.72	1.226	3.48	1.238	0.573	3.52	1.201	3.68	1.257	0.433
	DIY activities	3.49	1.103	3.59	1.229	0.003 *	3.59	1.189	3.59	1.133	0.345
	Featured itineraries and activities	3.53	1.063	3.36	1.242	0.002 *	3.55	1.091	3.40	1.174	0.213
Service facility	Quality of food and accommodation facilities	3.68	1.222	3.44	1.097	0.038	3.54	1.212	3.61	1.152	0.460
	Convenience of travel information	3.70	1.187	3.55	1.191	0.810	3.59	1.197	3.67	1.185	0.638
	Lighting equipment quality	3.57	1.097	3.51	1.169	0.037	3.51	1.186	3.57	1.084	0.041
	Number of public toilets	3.69	1.165	3.55	1.237	0.134	3.60	1.124	3.65	1.246	0.058
Space load	Parking spaces	3.67	1.118	3.47	1.209	0.038	3.55	1.200	3.61	1.132	0.092
	Convenient transportation	3.75	1.204	3.58	1.123	0.334	3.62	1.249	3.72	1.118	0.010 *
	Close attractions	3.67	1.179	3.59	1.227	0.064	3.64	1.196	3.63	1.202	0.558
	Spacious	3.80	1.027	3.49	1.235	0.000 *	3.62	1.101	3.71	1.145	0.515
Reputation	Media promotion	3.64	1.241	3.42	1.216	0.328	3.56	1.234	3.54	1.236	0.763
	Online marketing	3.54	1.106	3.65	1.122	0.218	3.60	1.119	3.58	1.110	0.448
	Recommended by relatives and friends	3.53	1.136	3.42	1.120	0.797	3.55	1.071	3.45	1.169	0.059
	School cooperation	3.55	1.202	3.48	1.150	0.364	3.49	1.178	3.54	1.184	0.916
Travel information and safety	Improve medical equipment	3.65	1.067	3.45	1.162	0.092	3.57	1.115	3.57	1.109	0.968
	Signs and explanations	3.67	1.041	3.51	1.201	0.000 *	3.64	1.033	3.58	1.164	0.011 *
	Community Association and Service Center Consulting Service	3.64	1.122	3.44	1.303	0.001 *	3.67	1.204	3.48	1.197	0.727
Price	Attraction tickets	3.56	1.139	3.44	1.226	0.188	3.47	1.138	3.54	1.202	0.270
	Travel vouchers can be used to offset daily consumption	3.63	1.161	3.58	1.085	0.323	3.57	1.133	3.63	1.129	0.789
	Many attractions without tickets	3.65	1.142	3.41	1.203	0.268	3.61	1.196	3.51	1.156	0.332
	Rental price of vehicles	3.65	1.186	3.47	1.215	0.409	3.55	1.223	3.59	1.185	0.747

* *p* < 0.01.

**Table 3 ijerph-18-11989-t003:** Analysis of cognitive differences in physical and mental health with different background variables.

Secondary Facets	Issue	Male		Female		*p*	Resident		Tourist		*p*
		M		SD			M		SD		
Inner feeling	Increase life satisfaction	2.45	0.972	2.52	0.822	0.000 *	2.47	0.918	2.49	0.911	0.874
	Full of enthusiasm	2.49	0.921	2.59	0.878	0.194	2.56	0.920	2.52	0.894	0.654
	Increase the efficiency of life skills	2.50	0.914	2.58	0.847	0.054	2.54	0.906	2.54	0.875	0.481
Mental state	Relieve headaches or overhead pressure	2.44	0.901	2.64	0.736	0.000 *	2.55	0.906	2.52	0.797	0.009 *
	Reduce backache problem	2.28	0.889	2.44	0.767	0.006 *	2.32	0.856	2.38	0.835	0.745
	No more insomnia	2.30	0.873	2.47	0.814	0.110	2.36	0.884	2.40	0.833	0.250
Life attitude and health	No stomachache and indigestion	2.19	0.859	2.42	0.821	0.984	2.26	0.883	2.33	0.831	0.435
	Restore appetite	2.27	0.924	2.57	0.855	0.071	2.35	0.910	2.40	0.899	0.819
	No longer anxious, losing your temper	2.38	0.966	2.54	0.862	0.004 *	2.42	0.918	2.47	0.934	0.662

* *p* < 0.01.

**Table 4 ijerph-18-11989-t004:** Analysis of cognitive differences in re-travel willingness with different background variables.

Issue	Male		Female		*p*	Resident		Tourist		*p*
	M		SD			M		SD		
Willing to continue to live or travel due to activities	2.64	0.945	2.67	0.905	0.277	2.68	0.960	2.64	0.906	0.336
Share living or travel experience	2.50	0.945	2.62	0.936	0.709	2.59	1.028	2.52	0.878	0.001 *
Willing to revisit the place	2.84	1.032	2.87	0.938	0.009 *	2.89	1.018	2.83	0.976	0.592

* *p* < 0.01.

**Table 5 ijerph-18-11989-t005:** Analysis of cognitive differences in leisure satisfaction, physical and mental health, and willingness to travel again with different background variables.

Facets		Leisure Satisfaction	Physical and Mental Health	Revisit Intention
Leisure satisfaction	Natural environment and landscape	0.910 **	−0.051	−0.200 **
	History, culture, and humanities	0.915 **	−0.065	−0.166 **
	Featured products	0.901 **	−0.058	−0.124 **
	Service facility	0.912 **	−0.008	−0.081 *
	Space load	0.916 **	−0.029	−0.183 **
	Reputation	0.920 **	−0.060	−0.191 **
	Travel information and safety	0.854 **	−0.017	−0.123 **
	Price	0.914 **	−0.042	−0.184 **
Physical and mental health	Inner feeling	−0.085 *	0.940 **	0.721 **
	Mental state	−0.036	0.979 **	0.636 **
	Attitude to life and health	−0.010	0.960 **	0.570 **
Revisit intention	Willing to continue to live or travel due to activities	−0.187 **	0.638 **	0.926 **
	Share living or travel experience	−0.181 **	0.677 **	0.931 **
	Willing to revisit the place	−0.113 **	0.525 **	0.883 **
Leisure satisfaction		1	−0.046	−0.175 **
Physical and mental health		−0.046	1	0.670 **
Revisit intention		−0.175 **	0.670 **	1

* *p* < 0.05 ** *p* < 0.01.

## Data Availability

Not applicable.

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
