# Peer review of "Can the Development of Religious and Cultural Tourism Build a Sustainable and Friendly Life and Leisure Environment for the Elderly and Promote Physical and Mental Health?"

_ijerph, 2021, doi:10.3390/ijerph182211989_

Round 1

Reviewer 1 Report

The argument is sometimes difficult to follow because the English is imperfect. Revision by an English-speaker is required. But the article appears original. I know of no other work with a comparable theme. The findings are convincing and can provide valuable advice to policymakers, though some comparative studies in other places would be helpful to establish to what extent the findings can be generalized. 

In my view this is an original and interesting article. I know of no other work that explores influences upon the physical and mental health and personal satisfaction of older people from these perspectives. The paper is clearly argued and well-focussed upon the paper should be revised by an experienced English speaker. The methodology appears sound and the conclusions are consistent with the evidence and arguments presented. I can see no ethical problems. The paper raises very interesting issues which it would be good to see replicated in research elsewhere in the world. He authors provide valuable advice for governments of policies to be adopted in order to maximize the health, satisfaction and contributions to their communities of older people.

Author Response

Dear reviewers:
Thank you for your suggestion and approval.
We try our best to finish the article retouching within a limited time.
We hope that the revised manuscript will also gain your approval.
Best regards, 

Reviewer 2 Report

Dear Authors,

In my opinion, this study contains valuable findings for the construction of living and leisure environments to improve the physical and mental functions of the elderly. However, I found some questions in the paper, and I would like to ask you to consider correcting them.

Introduction

On page 3, line 48, it says "Looking forward to using the research results to provide a reference for the government to make decisions about the physical and mental health care of the elderly and the development of cultural resources during the pandemic.” However, during a pandemic, travel and tourism tend to be restricted in order to prevent the spread of infection, making it difficult to utilize the results of this study. For this reason, I suggest that the argument be revised to say that it is necessary to build a better living and leisure environment in preparation for the post-pandemic period.

 Analysis and discussion

This study is being conducted under the special environment of a pandemic caused by Covid-19. There is no consideration of the impact of this special environment on the results of the study. In addition, I think that the consideration of the special environment under the Covid-19 pandemic should be added to the "Literature discussion".

Table

Some parts of Table 2 have not been translated into English, please correct them.

Author Response

Dear Authors,

In my opinion, this study contains valuable findings for the construction of living and leisure environments to improve the physical and mental functions of the elderly. However, I found some questions in the paper, and I would like to ask you to consider correcting them.

Introduction

On page 3, line 48, it says "Looking forward to using the research results to provide a reference for the government to make decisions about the physical and mental health care of the elderly and the development of cultural resources during the pandemic.” However, during a pandemic, travel and tourism tend to be restricted in order to prevent the spread of infection, making it difficult to utilize the results of this study. For this reason, I suggest that the argument be revised to say that it is necessary to build a better living and leisure environment in preparation for the post-pandemic period.
Dear reviewers:
Thank you for your suggestion and approval.
Our narrative in this section has been modified. As shown in lines 142-145.

 Analysis and discussion

This study is being conducted under the special environment of a pandemic caused by Covid-19. There is no consideration of the impact of this special environment on the results of the study. In addition, I think that the consideration of the special environment under the Covid-19 pandemic should be added to the "Literature discussion".
Dear reviewers:
Thank you for your suggestion and approval.
We have added relevant discussions in the chapters of literature discussion. As shown in lines 156-163.

Table

Some parts of Table 2 have not been translated into English, please correct them.
Dear reviewers:
Thank you for your suggestion and approval.
We have revised the Chinese description in the form. As shown in table 2.

Dear reviewers:
Thank you again for your suggestions and approval.
I believe that with your suggestions, the manuscript will be clearer and more complete.